# Supporting disengaged children and young people living with diabetes to self-care: a qualitative study in a socially disadvantaged and ethnically diverse urban area

Darren Sharpe  ,[1] Mohsen Rajabi,[2] Angela Harden,[3] Abdul Rehman Moodambail,[4] Vaseem Hakeem[5]

[1]Institute for Connected Communities (ICC), University of East London, London, UK
[2]University of Tehran, Tehran, Iran
[3]Centre for Maternal and Child Health Research, School of Health Sciences, City University of London, London, UK
[4]Department of Paediatrics, Newham University Hospital, Barts Health NHS Trust, London, UK
[5]Royal Free London NHS Foundation Trust, Barnet and Chase Farm Hospitals NHS Trust, London, UK

**Correspondence to**
Dr Darren Sharpe;
d.sharpe@uel.ac.uk

## ABSTRACT

**Objective** To explore how to enhance services to support the self-care of children and young people (CYP) clinically considered 'disengaged' by diabetes services.

**Design** Qualitative study.

**Setting** Two diabetes clinics in an ethnically diverse and socially disadvantaged urban area in the UK. Eligible participants were CYP living with type 1 or type 2 diabetes aged between 10 and 25 years who did not attend their last annual hospital appointment.

**Participants** 22 CYP (14 female and 8 male) aged between 10 and 19 years old took part. The sample was diverse in terms of ethnicity, age at diagnosis, family composition and presence of diabetes among other family members.

**Data collection** Semistructured interviews.

**Data analysis** Data were analysed thematically.

**Results** Analysis of participant accounts confirmed the crucial importance of non-medicalised care in CYP diabetes care. A life plan was considered as important to participants as a health plan. Participants valued the holistic support provided by friends, family members and school teachers. However, they found structural barriers in their health and educational pathways as well as disparities in the quality of support at critical moments along the life course. They actively tried to maximise their well-being by balancing life priorities against diabetes priorities. Combined, these features could undermine participants engagement with health services where personal strategies were often held back or edited out of clinical appointments in fear of condemnation.

**Conclusion** We demonstrate why diabetes health teams need to appreciate the conflicting pressures experienced by CYP and to coproduce more nuanced health plans for addressing their concerns regarding identity and risk taking behaviours in the context of their life-worlds. Exploring these issues and identifying ways to better support CYP to address them more proactively should reduce disengagement and set realistic health outcomes that make best use of medical resources.

## INTRODUCTION

Appointments can feel a bit intimidating – especially when there are five or six people in the room at the time. I do not like feeling judged or stared at. We should be able to have an open discussion about diabetes in relation to my risk-taking behaviour. (Participant 17, aged 13)

Diabetes self-management in children and young people (CYP) is a concern because of the assumption that adoption of diabetes self-care behaviours will lead to improved metabolic control of diabetes and will reduce the risk of complications in adulthood. Diabetes self-care includes a range of activities (eg, self-monitoring of blood glucose, eating a low-saturated-fat diet and checking one's feet), and it is now well established that these different components do not correlate highly.[1] In spite of structured education programmes and regular health promotion messages made by health professionals, researchers

### Strengths and limitations of this study

► Children and young people (CYP) who are usually marginalised and rarely heard in research were successfully recruited in this study including those from minority ethnic groups and those identified as 'disengaged' from health services.
► Diversity could have been increased further through recruitment of greater numbers of CYP with type 2 diabetes.
► The in-depth interviews were scheduled at times and locations chosen by the young person which helped to avoid replicating the power dynamics experienced in clinical appointments.
► The in-depth interview format was codesigned with a patient and public involvement (PPI) group, which supported CYP to talk openly on matters that concerned them most about diabetes self-care.
► The thematic framework was informed by both previous research and inquiry workshops with CYP led by the PPI group.

BMJ

and charities, we seldom hear the voices of CYP living with diabetes. We do not hear how they approach food and exercise alongside navigating the physiological and psychological changes consistent with growing up, or how they get their voices heard within health services in which we see a culture that is arguably dominated by paternalism.[2–5] The experience of being 'silenced' can often be compounded by parents, who do not give power away for putting their son's/daughter's healthcare plan into practice. Thus, for CYP living with diabetes, taking ownership of their self-care is far more complex than adherence to a health plan; they must negotiate and navigate a number of different relationships and contexts.

Research highlights the negative long-term health outcomes for members of the public who disengage from health services.[6–8] According to the UK National Health Service (NHS) 'disengagement' is defined when a CYP, or carer does not respond to requests from health professionals.[9] Behaviours of disengagement are usually cumulative and may include: disregarding health appointments; not having a general practitioner (GP); not being home for professional visits; not allowing professionals into the home; agreeing to take action but never doing it; hostile behaviour towards professionals; manipulative behaviour resulting in no healthcare; actively avoiding contact with professionals and attendance at urgent care centres, accident and emergency departments but not waiting to be seen/taking own discharge.[9] Elders *et al*[10] argue that people most likely to disengage are characteristically, 'young; are from more deprived areas; are more commonly anxious and depressed; have higher Glycated hemoglobin test (HbA1c) values; and are more frequently male' (p115).

Much of the medical literature on non-attendance in diabetes points to significantly higher HbA1c results among so-called 'defaulters' used as an example of the benefits of clinic attendance.[11] In England, there is evidence that young people miss more scheduled medical appointments of all kinds than other age groups.[12 13] Indeed, for younger patients, the transition from paediatric to adult diabetes clinics, is highlighted as a critical moment when young people drop out of the system.[14] However, reviews of the existing literature show a limited number of studies that have assessed the reasons behind clinic non-attendance, from the young patients' perspectives.[15 16] Understanding the challenges of CYP living with diabetes, from their perspective, especially in areas where a disproportionate burden of cases in diabetes falls on those from ethnic minority groups is needed. Most of the existing research does not focus on diverse groups of CYP or those from socially disadvantaged groups. So, this is about learning from those marginalised and/or not seen or heard voices in existing research.[15–17] This is an important first step to help shape the right diabetes care, at the right time, for CYP who live in communities that experience high deprivation and health inequalities.

This study was undertaken in response to a whole system call to improve accessible care for CYP with type 1 or type 2 diabetes living in London by understanding the barriers and identifying solutions to increase self-management. We focused on two boroughs within London, one of which has the highest local prevalence of type 2 diabetes in those between 16 and 25 years in the UK.[18]

Through qualitative methods this study seeks to deepen understanding into the barriers and drivers behind disengagement from services for CYP from socially deprived and ethnically diverse communities since these populations are most at risk of experiencing disparities in health provision and outcomes resulting from structural barriers. We examine how these barriers can be overcome and analyse CYPs independent self-care practices to explore what diabetes services can do to optimise safe self-care among this group.

## METHODS
### Design

In-depth interviews with CYP with type 1 or type 2 diabetes were used to assess their understanding and experiences of attending clinics and living with diabetes. The study was part of a larger programme of work undertaken between May 2016 to May 2018 focused on codesigning diabetes services for CYP. This broader programme included a systematic review of reviews, inquiry workshops and the development and evaluation of a young commissioner model in which young people with diabetes worked alongside adults to commission diabetes services.[19] The study followed steps recommended by the Standards for reporting qualitative resear (SRQR) guideline.[20]

### Patient and public involvement

The study was led by a team of adult researchers with the help of a group of young coinvestigators aged between 16 and 25 living with type 1 or 2 diabetes. The young co-investigators reviewed all study documentation including the study protocol, participant information sheets and the interview topic guide but they were not involved in the recruitment to and conduct of the study. Some members of this group also took on young commissioner roles within the larger programme in which this qualitative study is nested. The young coinvestigators met regularly over the study period (at least monthly) and received training in Diabetes 101, public speaking, workshop facilitation and commissioning public services. Findings of the overall programme were fed back to study participants via research briefings, a webinar series and public engagement events with targeted schools and diabetes networks attended by study participants and health professionals.

### Sampling and recruitment

We used a non-probability sampling strategy to identify, from two diabetes clinic registers, CYP categorised as 'disengaged' by their clinician as per the NHS Policy of Engagement and Disengagement with services definition.[9]

**Table 1** Characteristics of interview participants

| | N | | N |
|---|---|---|---|
| Gender | | No of siblings | |
| Male | 8 | 0 | 3 |
| Female | 14 | 1 | 12 |
| Age | | 2 | 3 |
| 10–13 | 9 | 3 | 2 |
| 14–17 | 4 | >3 | 2 |
| 18 and over | 9 | Position in family | |
| Ethnicity | | Eldest | 8 |
| Black African | 2 | Second eldest | 4 |
| Black Caribbean | 2 | Second youngest | 1 |
| Dual Heritage | 4 | Youngest | 6 |
| South Asian | 7 | Only child | 3 |
| White British | 6 | Adults in the household | |
| White other | 1 | Mother only | 4 |
| Diabetes type | | Mother and father | 15 |
| Type 1 | 20 | Other | 3 |
| Type 2 | 2 | | |
| Age at diagnosis | | Other family members with diabetes | |
| 0–4 | 8 | None | 12 |
| 5–9 | 5 | One other | 4 |
| 10–15 | 9 | More than one | 6 |

Potential participants were first approached by a member of the healthcare team, followed by a screening call from the research team. If the CYP were eligible (ie, they were aged between 10 and 25; living with type 1 or type 2 diabetes; living in North or East London; were pre or post a relevant transition such as moving from primary to secondary schools or moving to adult services; and did not attend their last annual hospital appointment) they were invited to take part in an interview.

A total of 22 CYP aged 10–19 years took part in the study out of 47 approached by the research team. Just over half of the sample were female and there was diversity in terms of age, ethnicity, age at diagnosis with diabetes, family composition, and presence of diabetes among other family members (table 1). All but two of the participants were living with type 1 diabetes.

## Data collection

Semistructured interviews were conducted to explore the personal journeys and lifestyles of CYP living with diabetes using a topic guide. Participants were asked:

► To describe their journey from diagnosis to now, including how they got diagnosed, how they felt, and what they would like to be different.
► What worked really well and what made life harder for participants that could be changed.

► How they managed their condition and what the health service (GPs, nurses and doctors) and other organisations (schools, youth centres and sports clubs) could do to more or less to positively impact on their life.
► Their future vision for diabetes services for CYP.

The majority of interviews took place in participant's homes (N=15) with five taking place at the university campus, one over the telephone and one at the participant's school. In half of the interviews (N=11) another family member was also present, most frequently the mother (N=7). Interviews were digitally recorded and transcribed.

## Data analysis

We analysed interview data using thematic analysis to find patterns of meaning that helped to explain CYPs 'disengagement' covering three focal points.[21] The three areas included their diabetes biography, social networks and future hopes and aspirations. Analysis involved repeated readings of the transcripts to gain familiarity with the content, the use of coding to identify recurring, similar and contrasting content, and the collapsing of codes into central themes by three members of the research team. Data validation was achieved by double coding a proportion of the transcripts followed by bringing all the coded transcripts into calibration meetings where we discussed and debated the constitution of each emergent theme and subtheme. Our coding scheme was also informed by insights from the systematic review of reviews and inquiry workshops with CYP led by our young coinvestigators from the wider programme of work and from health professionals and commissioners' insights from practice who formed part of the project task group.

Informed written consent was sought and received from all participants. A participant information sheet and consent form were sent to all potential participants at least 48 hours before a scheduled interview to allow time for them to consider their participation and ask any questions. On the day of the interview, participants were given a further opportunity to ask questions before signing a consent form. For potential participants aged 10–15 signed consent/permission was sought from their parent/legal guardian for their son/daughter to be approached to take part in the study. These younger participants were given information about the research project and were invited to sign an assent form following permission from their parent/carer. Findings and quotes in the report are pseudo-anonymised to minimise the risk of identifying participants.

## RESULTS

This section illustrates where and how participants negotiate and navigate the pressures from health services, education, home and social networks as part of their diabetes self-care. To recap, 47 potential interviewees agreed with their adult gatekeeper to be contacted by

the research team out of which 22 were successfully interviewed. Potential interviewees dropped out of the study due to episodes of homelessness, changing school diaries, and failure to show up to agreed interview appointments. The interviewed participants identified a range of issues they faced in living with diabetes including balancing the management of their diabetes with other aspects of their lives (eg, school, college or university, sports and hobbies), problems with maintaining glycaemic control (eg, difficulties carb counting, not liking healthy foods) and problems encountered within diabetes clinics (eg, poor relationships with clinic staff). The following themes were identified from participant interviews.

## Stigmatised status
Participants' shared common stories of how their peer groups behave, look and think. Accounts of peer group norms revealed how participants positioned themselves at the centre and/or margins of such norms rather than the actual perceptions and behaviour of their peers. Participants said:

> If I was to walk on the street, no one could actually tell that I had diabetes, unless I actually do something … for example a blood test… (P7, aged 15, T1).

Now at age 15, I have realised that diabetes doesn't make me different from anybody else (P10, aged 15, T1).

The relationship with self and others is complex and can positively and negatively impact on self-care. One participant explains:

I was sixteen and, on the bus, and I just didn't feel well, and I did my injection and the woman [passenger] said, 'oh my god', and I was, like, sorry yeah? And they're like, 'you know it's not good to take drugs on the bus?' And I'm like, 'excuse me', and my friend just started laughing like. Cos we couldn't believe it and I'm like 'it's not drugs. I'm diabetic' (P16, aged 17, T1).

Being a young person with diabetes was not the only marker of difference. Participants identified as belonging to ethnic minority groups (or linguistic groups), overlaid with intra group differences based on their gender, as well as lifestyle differences bought on by living with diabetes. Our analysis has shown that the perception of social stigma (eg, disapproval of a person based on perceived social characteristics) is a common feature negotiated in how participants experience building a stable social identity[22 23] while managing a chronic disease.[24 25]

For street-involved participants, the need to not appear vulnerable and to belong is often the reason given to why they might deviate from their health plans in order to navigate territorial stigmatisation of their identity and to increase the likelihood of inclusion by peers and in wider society. Ironically, participants reported feeling a greater sense of well-being, feeling safe and feeling valued in ethnically and/or religiously defined spaces than inside the clinic room, where they felt scrutinised and compartmentalised.

## Faith-based identity
Participants who identified as Christian or Muslim reported that their belief system played a significant role in their self-care, which could have both positive and negative consequences for the management of their diabetes. For example, participants gave accounts where religious observance helped them with their self-care. One participant said:

> As a Muslim you can't drink, [and I] don't smoke anyway, but there are rules like that, and, like, I do go out with friends but it's [drinking and smoking], not something I majorly think about (P13, aged 18, T1).

In contrast, two participants said:

> [Ramadan] is a sacred month for us. So, yeah, I want to be part of that sacred thing as well, so that's why I feel upset when I can't do it as well (P7, aged 15, T1).

> I had a DKA [diabetic ketoacidosis) over Ramadan. My mother was away and I was at home with my older sister. I wanted to experience fasting and the feast at the end. However, I ended up in hospital (P11, aged 18, T2).

Central to most, but not all, the participants' accounts was the importance they attached to their faith-based identities and how normative practices shared across their faith communities have been performed and, on occasion, have resulted in positive and negative self-care. Ramadan is an exemplifier of one such practice that can serve to resource identity but can also undermine healthy behaviours, so too are many other festivities and cultural events that require a break from routine eating patterns. Clinical teams should acknowledge the cultural spaces in which CYP inhabit and expect deviations from idealised medical plans.

## Supporting a CYP with diabetes in education
In the participants accounts of performing self-care their educational experiences were central features, with a specific focus on how teachers act as proxy healthcare workers. Participants said;

> …my tutor helps me a lot cos I have to test at the right time, and sometimes I forget, so my tutor is there to help me if I'm doing it right or I'm doing it wrong (P3, aged 11, T2).

> At school I use, like, a special room. I would disappear into it [to do injections].

> (P9, aged 15, T1)

> They [school] have an insulin register. If anybody is missing on the register, they go and look for them and remind them, have you eaten? (P7, aged 15).

Participants highlighted how policies and procedures in schools in poor communities, have helped to support them in their self-care. However, participants' accounts show challenges to self-care when transitioning between

schools and then onto college and University. Participants mentioned:

Going from primary to secondary school and then college is difficult in relation to the information and knowledge they [educators] have about my diabetes. …you receive less and less help and therefore educators know less and less about diabetes. (P11, aged 18, T2)

I experienced problems moving away from home to university and trying to sort out my studies, prescriptions and doctor appointments. (P2, aged 18, T1)

Participants highlighted the positive and negative features of school-based diabetes support,[26] with teachers playing a crucial role.[27] However, participants also observed a lack of training opportunities in schools to help teachers to improve their knowledge of diabetes care.[28–31] The overly reported challenge has been in transitions, when ironically support tends to tail off.[32] What is missing is a coordinated approach led by the health team in preventing the marginalisation of the CYPs health plans especially during stressful life events in their educational journeys, which result in high or imbalanced sugar levels leading to hospitalisation. In practice, participants have tended to figure things out for themselves when the healthcare team could help in a proactive way to produce a smooth transition.

Friendship networks, and not the healthcare team—formed a conceptual bridge between home and school, which was felt to be especially important to CYP with diabetes.[33 34] Peers are a central factor in a child's socialisation whether they have diabetes or not, but there are few studies addressing the role of friends among CYP in the management of the disease. A selection of accounts illustrates the type of support provided by friends in the self-care of participants in this study. Participants said:

I collapsed in the middle of the playground and he [best friend] was taught by my mum what to do. So, then he called my mum, who told him to call 999. (P7, aged 15, T1)

…when I know I'm going out for a drink with friends, I make sure that I have my meal and my sugar levels are good. When you're drinking, obviously I don't get in to a state where I don't know what I'm doing like. I'm scared to be, like, overly drunk and waking up in a diabetic coma or something. (P19, aged 18, T1)

Participants' accounts reveal how they use friendship networks as a source of support in their self-care, as argued by Salamon et al.[35] Yet still, social isolation and loneliness effects CYP and a few of the participant's experienced weak friendship networks and did not know other CYP in their cohort living with diabetes to relate too. Participants remark:

I think as a child you want to really open up and talk to someone that understands what you're going through, understands the injections, and the needles.

I think when I first got diabetes, I didn't know anyone with it so your kind of like, you can't talk about it with anybody. (P16, aged 17, T1)

I guess I don't get any real support. I have a couple of friends on Facebook, and we basically help each other through anything. (P10, aged 15, T1)

On the surface, healthcare teams appear absent in helping to educate friends or lack knowledge of the support provided by friends for CYP in their care. The home environment was important to understanding how participants made decisions affecting their self-care. The study takes as given the emotional consequences for CYP living in families undergoing sustained economic strain. CYP living in poverty are more likely to feel like a failure and have a sense of hopeless about their future than their more affluent peers. As a result, participants' choices around their self-care have often been made against the backdrop of health inequalities. In the face of economic hardships participants provided rich descriptions of how family members often serve as a crucial source of support in their self-care. Participants said;

I have lived with diabetes for the last thirteen years. It hasn't always been under control. It was when my mother took responsibility for me that my diabetes was more controlled, more freedom means less control. (P11, aged 18, T2)

I feel like because they [older family members] were like born with it [diabetes], they don't really talk to us about it, you know, how it's affected them or what they do. (P4, aged 12, T1)

I go to the gym with my auntie. It's like a peace of mind away from home. They have the little TV screens in front of each activity and on the cross trainer and the treadmill, and on the steps and things. So, I can just watch TV (P8, aged 18).

Seldom do studies capture the schooling experience in poor communities and how family circumstances reduce parental ability for active involvement in their child's self-care in school, especially from ethnic minority families. Evidence suggests that conscientiously caring for the carer helps CYP to reach their full potential in increasing resilience living with diabetes. This is reflected in: psychological models of assessment of care givers; understanding illness belief systems[36]; interdisciplinary working to help reduce family stress and anxiety[37]; including fathering a child with diabetes.[38] The increasing diversity experienced in urban areas such as London necessitates the cultural competency of the healthcare workers into the needs of families living in economic stress in order to provide the right support at the right time to care givers.

### Diabetes self-management

Too often, participants behaviour and attitudes have been wrongly characterised by clinicians as a marker of their disengagement from the service. The markers are hardly ever directly discussed by the healthcare team or

voluntarily disclosed in clinical appointments by CYP but loom large over the relationship. Participants said:

> At first, I didn't really take it [diabetes] that seriously, I suppose, but then when you realise it's going to affect you every day, then I started to take it more seriously. (P9, aged 15, T1)

> I try not to eat as much, so that my blood sugar level doesn't go so high and [it] saves me from having [to] reinject myself with more insulin. (P6, aged 13, T1)

> I'll eat what I want, I'll drink what I want and that's when my sugar levels started running high and I was just uncontrolled. So, I was missing appointments sometimes, like, oh, I'm just going to miss, I don't want to go. (P2, aged 18, T1)

> There are certain things you must cut back on. I used to do boxing, which was intense training, for an hour or so, so I think stuff like that I kind of left. I haven't been doing it as much. I feel it's quite a lot harder to do with diabetes. (P16, aged 17, T1)

In the latter case, no clinical support was provided to the participant to learn how to cope with high intensity exercise, in order to relieve the anger that so many of participants felt living with the condition. We see missed opportunity by healthcare teams to educate and support CYP to build liveable lives. Also, participants, highlight the inherent complications in implementing their health plan. Participants said;

> I'm constantly having to remember to do my blood test, and do my injections. Plus, with me, when I do my injection, after doing it, it does hurt. I get a bit frustrated and you don't want to do it afterwards. (P4, aged 12, T1)

> Most of the time, I find [doing injections] easy. Sometimes it's kind of annoys me that I must keep doing it and doing it, and sometimes it hurts, and, yeah, fear and perceptions of doing injections. (P2, aged 18, T1)

> You can't really tell I'm wearing a pump right now because everything is electronic blue tooth? I know that the pump and the meter will connect, automatically vibrate, so I will know the insulin will get delivered. Whereas, with the pen, I have to pick up my clothes, to open this, put a needle in the pen, do this, it's like a big long procedure before I actually give myself the injection. (P7, aged 15, T1)

Using technology in self-care is proposed to CYP as a way to promote better self-care; however, this benefit is not borne out in this study. Indeed, Balfe[29] highlights how little attention has been paid to the accounts of CYP of the reasons why they may experience difficulties with their diabetes technology. Participants said:

> I tried it [pump] for a while, but it's very painful. It can be an inconvenience practically. I was doing PE, for example, and I had it on, it would just be annoying because I couldn't play football or anything. (P20, aged 18, T1)

> I'm getting use to putting my carb count in the machine so it does the maths for me. I'm not really good at that because I like doing it mentally and, like, turn off the machine straight away. (P7, aged 15, T1)

The benefits of technology (eg, e-health) to aid self-care are inconclusive. Technological determinism is both embraced and resisted in the participants self-care narratives of type 1 participants. More examination needs to be undertaken exploring how today's CYP from poor and ethnically diverse communities navigate self-care, identity and intimacy in a digital world.

The five principle circumstances or markers that, so far as we have been able to observe, that drive the extent to which CYP with type, and a lesser degree type 2, are able to engage with health services and self-care include: (1) the probability or improbability of achieving normalcy when confronted with social stigma; (2) the easiness or difficulty in integrating faith-based practices into their daily routines; (3) the consistency or inconsistency in the quality of support provided in education; (4) the constancy or inconstancy of family and friends; and (5) the obstacles and practical challenges in balancing medicalised self-care practices in daily live.

## DISCUSSION

This study has systematically collected and considered CYPs perspective on diabetes services and factors influencing their self-care. Listening carefully to CYP, who had all been categorised as becoming disengaged with their diabetes care teams, has enabled the research team to amplify their voices and, through the wider programme of work in which this qualitative study is nested, informed commissioning guidance to improve diabetes services for CYP living in a poor and ethically dense communities in London. It is well recognised that CYP with long-term conditions may disengage from clinical services during adolescence and emerging adulthood. For CYP this can often contribute to poor health outcomes and for services a waste of medical resources.

Psychosocial needs and priorities of CYPs with type 1 diabetes and barriers to their engagement in educational settings[39] have been widely reviewed in the literature.[40 41] Most of these factors are centred around CYPs desire to lead a 'normal' life like their peers and must awkwardly adjust their language and behaviour to be considered as 'normal and healthy'. These factors may be less salient for those with type 2 diabetes as the management of their condition does not usually involve injections or pumps. Similarly, variations of the dynamics of engagement in self-care among black and ethnic minority groups have been discussed in the literature that emphasis peer pressure to meet subcultural expectations, which is shown to undermine health plans, and reinforce a lack of trust health professional.[42]

In contrast, the recent Commission for Race and Ethnicity Report in the UK[43 44] noted that the majority of all ethnic groups—which leave out the voices of CYP—reported positive experiences of access to healthcare and concludes that ethnic minority groups have better outcomes than the white population despite experiencing higher levels of deprivation. The factors at play are complex, and one way to account for the positive stories of self-care reported through this study—despite broken engagement with clinics—is through the cultural competencies narrated by participants in this study that have been applied to their self-care. For instance, Islamic law does not strictly forbid smoking, but it clearly stated that smoking is not good for health, and the health benefits of the Caribbean diet.

As illustrated in the findings, the personal, the situational, and the technical aspects of living with type 1 and type 2 diabetes problematise the simplistic and often unusual label given to black, Asian and minority ethnic group CYP who are considered to be 'disengaged', ignoring that participants are constantly trying to balance diabetes and life priorities often as a visible 'other'.[45] This paper stresses the role healthcare teams can perhaps play to address social determinants and/or sociological problems along the life course, to recognise CYP strengths and competencies, while they search for a stable identity and seek intimacy outside family relationships.[46]

At the centre of this discussion is the rebalancing of medicalised and non-medicalised self-care practices in CYP daily lives. The significance of this study is in arguing for a corrective emphasise, weighted evenly on the physiological and the psychosocial implications of diabetes for CYP from deprived and ethically diverse communities. Medicalised and non-medicalised self-care practices are not diametrically opposed in the participants accounts, but instead negotiated and navigated on an evenly momentarily or periodic basis. What this means for health service design and delivery is clear; rather than expecting CYP to adjust to health plans which are designed predominately by adults for CYP, health professionals should take into account the changing life circumstances and cultural priorities of CYP in order to increase meaningful engagement in health services.

Recognition should also be paid to the reality that self-care extends far beyond what is discussed in hospital appointments or determined by reading of CYPs blood sugar levels. The pressures of growing up in the 21st century weigh heavily on the self-care strategies adopted by the participants, and in turn affect how well CYP adhere to medicalised self-care plans (eg, doing carb counting, insulin injections, keeping to a healthy diet and regular exercise). The participant's accounts demonstrate how they are often preoccupied with the question of '*who I am*' and '*who I want to become*'. It goes without saying that the focus of most, if not all, of the participants has been on fitting in with their peer groups, thus often prioritising their social needs over their health needs.

Person-centred care planning is a promising way forward; however, despite policy being in place in England, practice is ad hoc. Form this context, the challenge for healthcare professionals is to create the right environment to design and deliver healthcare plans that model effective shared decision making grounded in a person-centred approach that takes account of CYP life circumstances. Engendering trust in the clinical relationship ought to be prioritised to allow for honest and frank discussions on lifestyles, behaviour and identity, which are not currently actively taking place in clinical appointments. Thus, more emphasis should be placed on integrating both medicalised and non-medicalised self-care techniques in CYP personalised plans, with a specific focus on the functioning of the CYP support network.

To conclude, this paper argues for non-medicalised self-care to gain parity with medicalised forms. In other words, a life plan is considered as important to the CYP as a health plan. We explore how both medicalised and non-medicalised forms of self-care often intersect in the daily lives of ethnically diverse CYP living in disadvantaged areas, and what this means for healthcare professionals. Diabetes teams need to appreciate the conflicting tensions experienced by CYP and to evolve better models for addressing their concerns regarding identity, risk and self-care in the context of their social setting and peer group. Exploring these issues and identifying ways to support CYP more effectively could reduce disengagement, improve health outcomes and make best use of healthcare resources.

**Acknowledgements** The authors would like to thank all participants and healthcare providers who contributed their time to share their perspectives with us. We would like to thank members of the research team Dr Emma Green and James Rudd. Most importantly we would like to thank our team of young co-investigators Sema Thasnim, Shandies Rose, Tahmid Alam and Khadija Miah.

**Contributors** All authors named contributed substantially to the document. DS and AH codesigned the study. DS and EG collected the data and DS, AH, EG and JR analysed the data. DS and MR wrote the draft manuscript. AH obtained the funding and provided support in editing the manuscript. DS, MR, AH, ARM and VH contributed to the design and critical review of the manuscript. All authors approved the final version. DS is the guarantor.

**Funding** This work was supported by the NHIR North Thames Collaborations for Leadership in Applied Health Research and Care.

**Disclaimer** The views expressed are those of the author(s) and not necessarily those of the NHS, the NIHR or the Department of Health and Social Care.

**Competing interests** None declared.

**Patient and public involvement** Patients and/or the public were involved in the design, or conduct, or reporting, or dissemination plans of this research. Refer to the Methods section for further details.

**Patient consent for publication** Consent obtained directly from patient(s)

**Ethics approval** Ethical approval was obtained by NRES Committee South East Coast-Surrey Bristol Research Ethics Committee Centre (REC reference: 15/LO/0903 IRAS project ID: 179878).

**Provenance and peer review** Not commissioned; externally peer reviewed.

**Data availability statement** No data are available. The interview audios and transcripts are not available to share outside of the research team.

**ORCID iD**

Darren Sharpe http://orcid.org/0000-0001-7418-4496

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
