## [Reviewer comments · BMJ Open]

ARTICLE DETAILS

TITLE (PROVISIONAL)	Supporting disengaged children and young people living with diabetes to self-care: A qualitative study in a socially disadvantaged and ethnically diverse urban area
AUTHORS	Sharpe, Darren; Rajabi, Mohsen; Harden, Angela; Moodambail, Abdul; Hakeem, Vaseem

VERSION 1 – REVIEW

REVIEWER	Goyder, Elizabeth ScHARR, University of Sheffield
REVIEW RETURNED	10-Jan-2021

GENERAL COMMENTS	This paper addresses an important and underexplored field with a focus on understanding the perspective of young people with diabetes who are perceived to be "disengaged". Some of the themes discussed in the Results section will certainly resonate with the experience of patients, their families and carers. However the research presented in this paper cannot be assessed or critically reviewed due to a lack of description of the study methods. The methods of data collection and analysis are not mentioned at all in the abstract. The Methods section very briefly mentions interviews and workshops and co-production with young people - and the bullet points on limitations/strengths mentions joint interviews with parents but none of these methods are described or justified. The Results section includes interview quotes but it is entirely unclear how the other elements of the methods (co-production, workshops, joint interviews) informed either the conduct or interpretation of the research. Similarly the Results does not describe the sample of children interviewed so it is impossible to judge the diversity or inclusivity of the recruitment or the characteristics of the sample of young people the data was collected from.. Overall the structure is inappropriate for a research article, starting with the study aim and the study conclusions rather than first presenting the rationale, then the aim, then the methods, then findings directly derived from the data collection described and then discussion/conclusions/implications for practice. More explanation of the ethical issues is needed, particularly informed consent and how it was decided whether parental consent was required and how the autonomy of children (of all ages) to given appropriate informed consent was respected. Minor issues - editing required for clarity of language and missing/inappropriate use of apostrophes throughout.
---

REVIEWER	Cooke, Debbie University of Surrey
REVIEW RETURNED	11-Feb-2021

GENERAL COMMENTS	This is an interesting and important qualitative study examining the reasons for children and young people's disengagement with diabetes services. Just a few points are recommended here for improvement:  1. The first two sentences of the introduction and the last paragraph would be better placed in the discussion/conclusions sections. 2. Information contained within the bullet points section on 'limitations' are not adequately discussed
---

REVIEWER	Agarwal, Shivani Albert Einstein Medical Center, diabetes, social determinants
REVIEW RETURNED	12-Feb-2021

GENERAL COMMENTS	This manuscript reports on results of a qualitative study to understand the non-medical needs of CYP with diabetes. While the overall message of diabetes providers accommodating non-medical needs for CYP with diabetes is important, I found the conclusions and discussion unfounded by the results. It was also unclear why you focused on disadvantaged populations and who your study population was given no participant characteristic reporting. Additionally, none of the results had to do with social disadvantage or social needs. Thus, the introduction and results felt disconnected from the title, abstract, and discussion. Following are some suggestions for how to improve the clarity and rationale of the manuscript: Introduction:  1. The first two paragraphs seem more appropriate for the discussion section. Before understanding the problem, which is presented in the 3rd and 4th paragraphs, there is interpretation and discussion of the issue. Would consider moving the initial paragraphs to lower down in the introduction, or to the discussion. 2. Page 2, lines 44-48: I find the following quote to be problematic: "However, reviews of the existing literature do not offer conclusive reasons for why and show that clinic-related factors behind non-attendance are rarely assessed, from the young patients perspectives [15,16]." There is qualitative research that evaluates various patient experiences that lead to poor pediatric to adult transition: Hilliard ME, Perlus JG, Clark LM, Haynie DL, Plotnick LP, Guttmann-Bauman I, Iannotti RJ. Perspectives from before and after the pediatric to adult care transition: a mixed-methods study in type 1 diabetes. Diabetes Care. 2014 Feb;37(2):346-54. doi: 10.2337/dc13-1346. Epub 2013 Oct 2. PMID: 24089544; PMCID: PMC3898755. Simms M, Baumann K, Monaghan M. Health Communication Experiences of Emerging Adults with Type 1 Diabetes. Clin Pract Pediatr Psychol. 2017 Dec;5(4):415-425. doi: 10.1037/cpp0000211. PMID: 29456906; PMCID: PMC5810591.
---

	3. The rationale for the study needs to be better detailed with regards to why you decided to focus on socially disadvantaged and ethnically diverse areas. There is no mention of this throughout the introduction, although this is heavily emphasized in the title and abstract. Methods: There is not enough explanation of the methodology as written.  1. Can you write more in the coding section to comment on reliability and validity of data coding? How many coders were used and how were discrepancies discussed? 2. Interview guide: Can you explain the domains of questions asked, the purpose of the questions, and give some example questions? Results: 1. What is the response rate of the study? 2. Please report the participant characteristics, especially given the emphasis on disadvantaged populations, need measures of socioeconomic status 3. The results detail typical competing priorities, but these are well documented in other literature, and do not go through any socioeconomic or social challenges related to ethnic minority status or neighborhood. The abstract is misleading given the results displayed. Discussion: While the discussion details important points, I do not find that the discussion ties back to the results presented. As mentioned prior, the results talk about non-medical needs of CYP, but they do not tie back to disengagement or social disadvantage as laid out in the abstract or the rationale of the study. In addition, there are many manuscripts on non-medical needs of CYP with diabetes, however these are largely excluded. I would include a more comprehensive literature review of competing priorities and psychological needs in CYP with diabetes and discuss your results in the context of the current literature. At this point in time, given the results that are exhibited, it is unclear how this manuscript adds to the existing literature on this topic. I would discuss differences between type 1 and type 2 diabetes needs, if you included both types of diabetes in this study. Their self-care needs are quite different and should be explained to readers.
--	---

VERSION 1 – AUTHOR RESPONSE

AUTHOR RESPONSE

Editor	Response
Please revise your Strengths and Limitations section to include limitations which relate specifically to the methods in your study. This section should contain five	Revised based on method only.

short bullet points, no longer than one sentence each, that relate specifically to the methods.

2) Patient and Public Involvement:

- We have implemented an additional requirement to all articles to include 'Patient and Public Involvement' statement within the main text of your main document. Please refer below for more information regarding this new instruction:

Authors must include a statement in the methods section of the manuscript under the sub-heading 'Patient and Public Involvement'.

This should provide a brief response to the following questions:

How was the development of the research question and outcome measures informed by patients' priorities, experience, and preferences?

How did you involve patients in the design of this study?

Were patients involved in the recruitment to and conduct of the study?

How will the results be disseminated to study participants?

For randomised controlled trials, was the burden of the intervention assessed by patients themselves?

Patient advisers should also be thanked in the contributorship statement/acknowledgements.

If patients and or public were not involved please state this.

A new section of Patient and Public Involvement has been added to the manuscript as follows:

"Patient and Public Involvement

The study was led by a team of adult researchers with the help of a group of nine young co-investigators aged between 16 and 25 living with Type 1 or 2 diabetes. The young co-investigators reviewed all study documentation including the study protocol, participant information sheets and the interview topic guide but they were not involved in the recruitment to and conduct of the study. Some members of this group also took on young commissioner roles within the larger programme in which this qualitative study is nested. The young co-investigators met regularly over the study period (at least monthly) and received training in Diabetes 101, public speaking, workshop facilitation and commissioning public services. Findings of the overall programme were fed back to study participants via research briefings, a webinar series and public engagement events with targeted schools and diabetes networks attended by study participants and health professionals."

Members of the PPI group have now been thanked in the acknowledgements section.

Reviewer 1

Responses

This paper addresses an important and underexplored field with a focus on understanding the perspective of

Thank you very much for taking the time to review our manuscript and for your supportive view and

young people with diabetes who are perceived to be "disengaged". Some of the themes discussed in the Results section will certainly resonate with the experience of patients, their families and carers.

constructive comments. We responded to the raised points below.

However the research presented in this paper cannot be assessed or critically reviewed due to a lack of description of the study methods. The methods of data collection and analysis are not mentioned at all in the abstract. The Methods section very briefly mentions interviews and workshops and co-production with young people - and the bullet points on limitations/strengths mentions joint interviews with parents but none of these methods are described or justified.

These points have now all been addressed:

*Methods of data collection and data analysis are now featured in the abstract

*The study of focus in this paper has now been clarified in the context of a wider programme of working featuring workshops and co-production elements (young commissioners) to make it clear that this paper focuses on the in-depth qualitative interviews conducted with young people. Reference is given to where we report other elements of the programme have been given.

*We have now clarified that no joint interviews were conducted. In half of the interviews parents were present in the interview but they themselves were not interviewed.

*The characteristics of participants have now been fully described (see new Table 1)

The Results section includes interview quotes but it is entirely unclear how the other elements of the methods (co-production, workshops, joint interviews) informed either the conduct or interpretation of the research. Similarly the Results does not describe the sample of children interviewed so it is impossible to judge the diversity or inclusivity of the recruitment or the characteristics of the sample of young people the data was collected from.

The manuscript now includes a fuller methods section which clearly spells out the data the manuscript reports on was gathered through semi-structured interviews and the workshops and co-production formed part of the wider programme activities and feed into the development of the coding system used in the coding and analysis of the interview transcripts.

Overall the structure is inappropriate for a research article, starting with the study aim and the study conclusions rather than first presenting the rationale, then the aim, then the methods, then findings directly derived from the data collection described and then discussion/conclusions/implications for practice.

Thank you for your comment. We revised abstract, introduction and discussion to redesign the structure of the manuscript.

More explanation of the ethical issues is needed, particularly informed consent and how it was decided whether parental consent was required and how the autonomy of children (of all ages) to give appropriate informed consent was respected.

This has been added to the ethics section as follows:

“Informed written consent was sought and received from all participants. A participant information sheet and consent form were sent to all potential participants at least 48 hours before a scheduled interview to allow time for them to consider their participation and ask any questions. On the day of the interview participants were given a further opportunity to ask questions before signing a consent form. For potential participants aged 10-15 signed consent/permission was sought from their parent/legal guardian for their son/daughter to be approached to take part in the study. These younger participants were given information about the research project and were invited to sign an assent form following permission from their parent/carer. Findings and quotes in the report are pseudo-anonymised to minimise the risk of identifying participants.”

Editing required for clarity of language and missing/inappropriate use of apostrophes throughout.

Thank you for your feedback. We did a thorough language check and made some small improvements/amendments. We hope that our amendments have made the manuscript easier to read.

Reviewer 2

Responses

This is an interesting and important qualitative study examining the reasons for children and young people's disengagement with diabetes services. Just a few points are recommended here for improvement.

Thank you for taking the time to review our manuscript and we are glad to hear that you found it interesting and important study. Please find our responses to the raised points below.

The first two sentences of the introduction and the last paragraph would be better placed in the discussion/conclusions sections.

Thank you for your suggestion. We can confirm that this section has now been moved down from the introduction section into the discussion section (see page 19).

Information contained within the bullet points section on 'limitations' are not adequately discussed within the manuscript.

We revised abstract to show more clearly what are the limitations of this study. The comment led us to also revise the bullets for the 'Strengths and limitations'.

Reviewer 3**Responses**

This manuscript reports on results of a qualitative study to understand the non-medical needs of CYP with diabetes. While the overall message of diabetes providers accommodating non-medical needs for CYP with diabetes is important, I found the conclusions and discussion unfounded by the results. It was also unclear why you focused on disadvantaged populations and who your study population was given no participant characteristic reporting. Additionally, none of the results had to do with social disadvantage or social needs. Thus, the introduction and results felt disconnected from the title, abstract, and discussion. Following are some suggestions for how to improve the clarity and rationale of the manuscript:

Thank you for taking the time to review our manuscript and for these supportive and constructive comments. We followed the suggestions and revised the introduction section and the result section accordingly. We have also responded to the raised points below.

Introduction:

1. The first two paragraphs seem more appropriate for the discussion section. Before understanding the problem, which is presented in the 3rd and 4th paragraphs, there is interpretation and discussion of the issue. Would consider moving the initial paragraphs to lower down in the introduction, or to the discussion.

Thank you for pointing this out. We can confirm that this section has now been moved down from the introduction section into the lower down in the introduction and discussion section (see pages 19).

Page 2, lines 44-48: I find the following quote to be problematic: "However, reviews of the existing literature do not offer conclusive reasons for why and show that clinic-related factors behind non-attendance are rarely assessed, from the young patients perspectives [15,16]."

Thank you for suggesting the references and input. We revised and added the recommended studies on page 5. The revised sentences is as follows:

There is qualitative research that evaluates various patient experiences that lead to poor pediatric to adult transition:

" However, reviews of the existing literature show a limited number of studies that have assessed the reasons behind clinic non-attendance, from the young patients' perspectives [15,16]."

Hilliard ME, Perlus JG, Clark LM, Haynie DL, Plotnick LP, Guttman-Bauman I, Iannotti RJ. Perspectives from before and after the pediatric to adult care transition: a mixed-methods study in type 1 diabetes. *Diabetes Care*. 2014 Feb;37(2):346-54. doi: 10.2337/dc13-1346. Epub 2013 Oct 2. PMID: 24089544; PMCID: PMC3898755.

Simms M, Baumann K, Monaghan M. Health Communication Experiences of Emerging Adults with Type 1 Diabetes. *Clin Pract Pediatr Psychol*. 2017 Dec;5(4):415-425. doi: 10.1037/cpp0000211. PMID: 29456906; PMCID: PMC5810591.

The rationale for the study needs to be better detailed with regards to why you decided to focus on socially disadvantaged and ethnically diverse areas. There is no mention of this throughout the introduction, although this is heavily emphasized in the title and abstract.

We have updated the manuscript. The rationale behind why we focus on 'disagreement among CYP in poor ethnically diverse communities is due to the lack of empirical research that tells their stories despite the high levels of diabetes in these areas and the fact that a disproportionate burden falls on those from ethnic minority groups. Plus, the majority of existing research does not focus on/can't tell whether they focus on diverse groups of young people or those from socially disadvantaged groups so this is about learning from those marginalised/not seen or heard in existing research. In discussion will need to compare our findings to those from existing research which focuses on (we suspect) CYP perspectives from more advantaged groups.

Methods:

There is not enough explanation of the methodology as written.

1. Can you write more in the coding section to comment on reliability and validity of data coding? How many coders were used and how were discrepancies discussed?

The manuscript has been updated with information on the methodology. Semi-structured interviews were conducted to explore the personal journeys and lifestyles of CYP living with diabetes using a topic guide. Participants were asked :

- to describe their journey from diagnosis to now, including how they got diagnosed, how they felt, and what they would like to be different;
- what worked really well and what made life harder for participants that could be changed;
- how they managed their condition and what the health service (GPs, nurses, and doctors) and other organisations (schools, youth centres, and sports clubs) could do to more or less to positively impact on their life; and
- their future vision for diabetes services for children and young people. ”

2. Interview guide: Can you explain the domains of questions asked, the purpose of the questions, and give some example questions?

Thank you for your comment. We added new headline "Data Collection" and explained the domains of questions, and some example of items. You can see page 8-9.

Results:

1. What is the response rate of the study?

We have revised the manuscript to reflect that from the 47 potential participants who said yes to take part in the study via their diabetes health worker only 22 participants were finally interviewed (see page 10).

2. Please report the participant characteristics, especially given the emphasis on disadvantaged populations, need measures of socioeconomic status.

Thank you for the comment. We added "Sampling and recruitment" headline and Table 1 to report on the participants characteristics. You can find this on page 7-8.

3. The results detail typical competing priorities, but these are well documented in other literature, and do not go through any socioeconomic or social challenges related to ethnic minority status or neighborhood. The abstract is misleading given the results displayed.

We have addressed and rebalanced this narrative throughout the manuscript. Whilst ethnic and deprivation are two important variables we consider the paper is on 'disengagement' and how certain voices are missing in research to better understand this phenomena. The study is not about poverty or 'race' per se. See pages 5-6 for full explanation.

Discussion:

While the discussion details important points, I do not find that the discussion ties back to the results presented. As mentioned prior, the results talk about non-medical needs of CYP, but they do not tie back to disengagement or social disadvantage as laid out in the abstract or the rationale of the study.

The discussion section has been updated to thread together the discussion of key elements of the results to better reflect Type 1 and 2 and nuances as it relates to the study question.

In addition, there are many manuscripts on non-medical needs of CYP with diabetes, however these are largely excluded. I would include a more comprehensive literature review of competing priorities and psychological needs in CYP with diabetes and discuss your results in the context of the current literature.

We have added some related studies on this topic (e.g. you can see ref no 41).

At this point in time, given the results that are exhibited, it is unclear how this manuscript adds to the existing literature on this topic.

We have revised introduction, results and discussion to show more clearly what our study adds to the existing literature.

I would discuss differences between type 1 and type 2 diabetes needs, if you included both types of diabetes in this study. Their self-care needs are quite different and should be explained to readers.

Thank you for the input. The suggested additional explanations are included now. We have highlighted the added text on pages 19-20.

VERSION 2 – REVIEW

REVIEWER	Cooke, Debbie University of Surrey
REVIEW RETURNED	06-Jul-2021

GENERAL COMMENTS	This is a valuable paper and I enjoyed reading it. The revisions have definitely improved the quality of the paper. I have one very minor recommendation I would suggest amending the first sentence to omit reference to “healthy lifestyles”, as follows: Diabetes self-management in children and young people (CYP) is a concern because of the assumption that adoption of diabetes self-care behaviours will lead to improved metabolic control of diabetes and will reduce the risk of complications in adulthood.
--